# Impact of Coronavirus Pandemic on Tuberculosis and Other Imported Diseases Screening among Migrant Minors in Spain

**DOI:** 10.3390/tropicalmed8010028

**Published:** 2022-12-29

**Authors:** Isabel Mellado-Sola, Paula Rodríguez-Molino, Elisa-Andrea Armas, Javier Nogueira López, Iker Falces-Romero, Cristina Calvo Rey, Carlos Grasa Lozano, María José Mellado, Milagros García López-Hortelano, Talía Sainz

**Affiliations:** 1General Pediatrics, Infectious and Tropical Diseases Department, Hospital La Paz, 28046 Madrid, Spain; 2La Paz Research Institute (IdiPAZ), 28029 Madrid, Spain; 3Faculty of Medicine, Universidad Autónoma de Madrid (UAM), 28049 Madrid, Spain; 4Centro de Investigación Biomédica en Red en Enfermedades Infecciosas (CIBERINFEC), Madrid, Spain; 5Quirón Salud San José, 28002 Madrid, Spain; 6Centro de salud de Cascastillo, 31310 Navarra, Spain; 7Department of Microbiology and Parasitology, La Paz University Hospital, 28046 Madrid, Spain; 8Red de Investigación Traslacional en Infectología Pediátrica (RITIP), Madrid, Spain

**Keywords:** COVID pandemic, unaccompanied minors, children, screening, tuberculosis, parasites, immunizations

## Abstract

Background: In recent decades, the increase in population movements has turned the focus to imported diseases. The COVID-19 pandemic has negatively impacted the access to health care systems, especially in highly vulnerable populations. We address the effects of the pandemic on the health screening of migrant unaccompanied minors (UM) in Spain. Method: Retrospective cross-sectional study including UM screened for imported diseases with a unified protocol at a pediatric reference unit for tropical and infectious diseases in Madrid, Spain. We compared the pre-pandemic (2018–2019) and post-pandemic periods (2020–2021). Results: A total of 192 minors were screened during the study period, with a drop in UM’s referral to our center in the post-pandemic years (140 in 2018–2019 vs. 52 in 2020–2021). Out of 192, 161 (83.9%) were diagnosed with at least one medical condition. The mean age was 16.8 years (SD 0.8) and 96.9% were males. Most cases were referred for a health exam; only 38% of children were symptomatic. Eosinophilia was present in 20.8%. The most common diagnosis were latent tuberculosis infection (LTBI) (72.9%), schistosomiasis (15.1%), toxocariasis (4.9%) and strongyloidiasis (4.9%). The prevalence of LTBI did not vary significantly (69.3% vs. 82.7%, *p* = 0.087). A total of 38% of the patients diagnosed with LTBI never started treatment or were lost to follow-up, as were two out of three patients with active pulmonary tuberculosis. Conclusions: In this series, the number of UM referred for health screening has dropped dramatically after the COVID pandemic, and two years after the beginning of the pandemic, access to care is still limited. Lost to follow-up rates are extremely high despite institutionalization. Specific resources, including multidisciplinary teams and accessible units are needed to improve diagnoses and linkage to care in this vulnerable population.

## 1. Introduction

The devastating effects of the COVID-19 pandemic on global health have been widely described. Reduced access to care and essential health care disruptions secondary to outbreaks have disproportionately affected the most disadvantaged populations, increasing the burden of poverty related diseases [1,2,3]. In 2021, the WHO registered for the first time in more than a decade, an increase in tuberculosis (TB) mortality [1] and in the coming years, the drop in case detection and subsequent treatment would probably lead to an increase in transmission and a rise in the number of cases. In children, BCG vaccination has decreased by an estimated 5% and up to 63% of children and adolescents under 15 years of age did not receive TB treatment globally [4]. The risk of TB progression is higher during childhood and adolescence, and failure to diagnose latent infection in these vulnerable populations may significantly increase the burden of disease.

Spain is one of the European countries with the most confirmed cases of SARS-CoV-2, only surpassed by France, Germany, the United Kingdom and Italy; it ranks 12th of countries with the most cases in the world, despite its small geographical area [5]. In Spain, the number of cases diagnosed with TB also decreased in the post-pandemic years. However, there has been an increase in cases with bilateral infiltrates probably caused by the delay in diagnosis [6].

Unaccompanied migrant minors (UM), children and adolescents that migrate without accompanying adults, represent one third of all asylum seekers in the European Union (EU) [7]. In 2015, the numbers peaked in the EU, with 92,000 minors registered as asylum applicants. Since then, there has been a generalized decreasing trend. In 2020, despite the Coronavirus outbreak, the number of UM seeking asylum reached 13,600, which is only a 4% lower than in 2019 [7]. This is a highly vulnerable population posing both legal and health care challenges [8,9,10,11,12,13] who concentrate in certain countries of the EU. Although Spain has not been traditionally among the preferred countries to request asylum—Germany, France or Italy were usually at the top of the list—the number of asylum seekers in Spain has grown progressively since 2015 in contrast with European statistics [7]. During 2022, the effect of the pandemic was already unnoticeable in the country, accounting for over 11,000 first-time asylum seekers, representing 14.8% of the EU total. The instability in regions such as Venezuela and Colombia is probably one of the reasons that has caused the registered shifts, and owing to the language and the cultural and historical links between Spain and Latin America. Due to its position in the western Mediterranean migration route, Spain is also a hot spot for migrants from Sub-Saharan and Northern Africa [12], the origin of many UM in the EU, right behind Afghanistan and Syria. 

The screening of imported diseases in UM is key to prevent transmission and long-term sequelae. However, there are many obstacles in their access to the health system (e.g., language and cultural differences, overcrowding and lack of resources in the refugee centers), which complicates their linkage to care and follow-up [8,14,15,16,17,18]. This is a concern, especially for the prevention of TB, which requires long treatments often not well tolerated, with frequent adherence issues. 

The aim of this study is to describe the effect of the COVID pandemic on the screening of UM at a Reference Pediatric Tropical Unit in Madrid, Spain. We describe the prevalence of tuberculosis and other infectious diseases, the diagnostic performance of a tailored protocol for the screening of imported diseases and the longitudinal follow-up. 

## 2. Methods

### 2.1. Study Design and Participants

A retrospective, observational study was performed at a National Reference Unit for Pediatric Tropical Diseases in Spain. All UM below 18 years of age referred for health control from primary care, the emergency room and/or reception centers for asylum seekers and refugees between January 2018 and December 2021 were included. Medical records were reviewed. All patients were managed according to a unified protocol to screen for imported diseases. Blood tests (complete blood count, biochemistry including renal and liver function) and serology for helminths, congenital and vaccine-preventable diseases were conducted. Three fecal samples were collected for stool concentration microscopy.

TB screening was carried out according to the protocol of the unit and the criteria of the clinician. Most PPD tests were performed in primary care centers before being referred to our unit, IGRAs were performed simultaneously or sequentially and in those with positive results, chest radiography was performed. In cases with suspected TB disease, samples of sputum and/or gastric aspirate were collected for culture and PCR. 

The research was reviewed and approved by the Ethical and Research Committee of Hospital Universitario La Paz (PI-3348). Due to its retrospective nature, no informed consent was required.

### 2.2. Definitions

The pre-pandemic period included two years prior to the pandemic (2018, 2019) and January–February 2020, whereas the pandemic years included March 2020 to December 2021.

Parasitism was defined when at least one parasite (pathogenic or not) was identified. Co-parasitism was considered when at least two parasites were found. We subsequently analyzed pathogenic and nonpathogenic parasites. Pathogenic parasites considered all helminths and pathogenic protozoa: *Giardia intestinalis*, *Cryptosporidium parvum* and *Entamoeba histolytica*. Parasites with a controversial pathogenic effect were categorized with nonpathogenic parasites: *Endolimax nana*, *Blastocystis* sp., *Entamoeba dispar/hystolitica* when not differentiated, *Entamoeba dispar*, *Entamoeba coli*, *Entamoeba hartmanni*, *Iodamoeba bütschlii* and *Dientamoeba fragilis* [19].

Anemia was defined according to WHO as hemoglobin (Hb) <12 g/dL [20]. Eosinophilia was defined when peripheral eosinophil count >500/microL [21]; relative eosinophilia when an absolute eosinophil count <500/microL but represented >5% of circulating leukocytes [18]. A peripheral leukocyte count above 11.000/mm^3^ defined leukocytosis [22], whereas increased total serum immunoglobulin E (IgE) levels when >100 U/mL [18]. Hypertransaminasemia was considered when AST > 50 U/L and/or ALT > 45 U/L [23]. Hypercholesterolemia was defined as total cholesterol >200 mg/dL or LDL-cholesterol >130 mg/dL [24].

Body mass index (BMI) was automatically calculated and considered low below 18.5 kg/m^2^ and elevated when >25 kg/m^2^ [25].

### 2.3. Diagnosis

For the detection of parasites, both direct and indirect methods were used. Direct techniques consisted of the examination of three stool samples taken on alternate days. Firstly, the Mini Parasep^®^ SF fecal parasite concentrator was used, and then optical microscopy looking for ova, cyst, larvae or parasites. Additionally, in patients coming from an endemic area and/or presenting with eosinophilia in which schistosomiasis was suspected, a urine sample for ova of *Schistosoma* was analyzed using Millipore Swinnex^®^ membrane filter-holders (25 mm) and Whatman Nuclepore^®^ membrane (10 μm). Indirect tools consisted of commercial serologic tests for *Strongyloides* (SciMedx microwell-ELISA^®^), *Toxocara* (NovaTec IgG-ELISA^®^) and *Schistosoma* (NovaTec IgG-ELISA^®^). ImmunoCAP Specific IgE-EIA tests (Thermo Fisher^®^) for *Ascaris lumbricoides* and *Anisakis simplex* were performed on an individual basis and usually on a second stage. Tuberculosis PCR was performed using GeneXpert^®^ Ultra (Cepheid^®^). The cut-off was established by the manufacturing company. 

### 2.4. Statistical Analysis

We used descriptive statistics to summarize demographic and clinical characteristics. Categorical variables were presented as absolute frequencies and percentages. Continuous variables were presented as the mean, with its standard deviation (SD). Categorical variables were compared using the Chi-square test, likelihood ratio test or Fisher’s Exact test, as appropriate. Continuous variables were compared by Student’s *t*-test or the Mann-Whitney–U test according to their distribution. A two-tailed *p*-value was set at 0.05 for all tests. All statistical analyses were performed using IBM-SPSS Statistics v.23. 

## 3. Results

### 3.1. Study Population

During the study period, 192 UM were referred to and screened at our Unit; 97% were male, with a mean age of 16.8 years (±0.8). The demographic and clinical characteristics of the study participants are summarized in Table 1. All participants except one came from Africa, Morocco being the country of origin of most of the adolescents (57%). A total of 143 (74.5%) youths were referred due to a positive tuberculin test (PPD) at arrival at any of the centers for asylum seekers, and another 20.8% for study due to eosinophilia >500/microL. Overall, only 73 patients (38%) were symptomatic. The most frequently referred symptoms were coughing (10%) and skin lesions (10%), followed by abdominal discomfort (5.7%). Diarrhea was extremely infrequent (2%). Clinical examination revealed cavities and/or oral abscesses in 54 patients (28.1%). Blood tests were performed in all patients; 20.8% presented with eosinophilia, but only five patients (2.6%) were diagnosed with anemia and 6 (3.1%) with hypercholesterolemia. Clinical characteristics and main findings during evaluation or laboratory test are summarized in Table 1, along with the main diagnosis. Latent tuberculosis infection (LTBI), diagnosed in 140 patients (72.9%), was the most frequent condition, followed by schistosomiasis (29 patients, 15.1%). Screening for parasites in the stool was performed in only 72 patients, with 54 (75%) presenting at least one parasite (Table 1).

### 3.2. Tuberculosis Diagnosis and Follow-Up

All patients diagnosed with LTBI but one (lost to follow up before the results of QuantiFERON test were available) were treated. The most common regimen was isoniazid and rifampicin combined treatment for three months (93.6%). Five (3.6%) received isoniazid monotherapy and one received rifampicin in monotherapy because of comorbidities and/or interactions. Regarding follow-up, out of 140 LTBI patients, 89 (63.6%) completed the treatment with good adherence and no adverse effects, and 4 (2.9%) reported side effects. Follow-up data were unavailable for 47 (33.6%) patients diagnosed with LTBI. The main reasons were that they were lost to follow-up after they left the center for asylum seekers or after transfer to another region. A total of 21 patients (15% of LTBI diagnosis) were only seen once, and therefore never initiated treatment. There were no differences in the number of patients lost to follow-up when comparing the pre- and post-pandemic periods. 

Three patients (1.6%) with abnormalities in imaging tests were diagnosed with active tuberculosis disease. All three were treated with four drugs. Only one maintained good adherence and tolerance until completion of treatment. The second case was lost to follow up during the first weeks of treatment, reporting good tolerance but regular adherence. The third one dropped out of the follow-up before the first control visit and therefore tolerance and adherence are unknown.

### 3.3. Other Infections and Immunization

Among other infections, active HBV infection stands out in 16 patients (8.3%) and past infection in 23 (12%). The most frequent parasites detected were *B. hominis* (62.5%) and *Schistosoma* spp. (44.7%), followed by *D. fragilis* (22.5%) and *G. lamblia* (12.5%). Treatment for parasitosis was prescribed in 34 patients; 65% (22 patients) reported good adherence to treatment and follow-up was lost in 32.4% (11 patients). 

Regarding immunizations, the serostatus for the different immunizations analyzed is summarized in Table 1. Most patients required immunizations, with 61.7% of participants that were not protected against HVB and 31.6% against measles.

### 3.4. Effect of the Pandemic

Over the study period, the number of UM referred for consultation decreased significantly (by 62.85%) after the first COVID-19 outbreak, compared to the two previous years. The drop was especially relevant during the first semester of 2020, as shown in Figure 1. Two years after the start of the pandemic, in December 2021, the numbers were still far from the pre-pandemic period. Comparing the demographic variables per period, we found a decrease in the proportion of participants arriving from Morocco after the pandemic outbreak (61.4% vs. 44.2%; *p* = 0.024). Although the absolute number decreased significantly, the rate of patients referred for study due to a positive PPD test was comparable between both periods (71.4% vs. 82.7%, *p* = 0.599). There were no significant differences in the number of patients lost to follow-up comparing the pre- and pandemic periods: 52 (38%) pre-pandemic and 23 (44%) post-pandemic. Interestingly the rate of participants with protective titers against measles decreased in the pandemic period (74.6% vs. 52.3%, *p* = 0.07).

## 4. Discussion

The results of this study show the impact of the COVID-19 pandemic on access to health among UM arriving to the EU, with a 63% decrease in referrals. Two years after the first outbreak, the numbers have not recovered, whereas according to the official statistics, the rate of asylum seekers in the region has grown again. Latent tuberculosis infection was extremely frequent, with a comparable prevalence in the pre- and post-pandemic periods (69.3% vs. 82.7%; *p* = 0.087). However, in absolute numbers, there was a decrease by half in the number of cases. These findings are in line with the 2021 WHO Global Report [26] that describes a drop in case notifications of new diagnosis worldwide [27,28]. The data, combined with the poor adherence and linkage to care observed, with 21% of patients not even starting treatment and around 40% of patients lost to follow-up over time, are extremely worrisome and endanger the control of transmission. Although those infected are usually asymptomatic, parasitosis was also common in this population, underlying the need for defining surveillance strategies, and optimizing screening and follow-up plans for this vulnerable population. 

Imported diseases are a public health concern inherent to population movements. Infectious diseases such as TB or parasitosis are extremely linked to poverty and the social determinants of health, and often pauci are asymptomatic for long periods of time. Screening for imported diseases in asylum seekers and migrants is both an opportunity for early diagnosis and treatment that can avoid future sequelae and a public health need. However, the need to adjust diagnostic tests to symptoms and local epidemiology and the scarce evidence available limits our ability to standardized protocols. In the evaluation of UM, the epidemiology at origin needs to be supplemented with the potential risk exposure during the migration process, which is often extremely long and arduous. Still, cultural and language barriers often impair a proper anamnesis (as described in this series), and fear and the sense of helplessness are difficult to overcome. The benefits of tuberculosis screening might be clear, but the optimal test to use may vary according to availability. A PPD skin test is cheaper and accessible, but requires two visits, and it may be misinterpreted because of BCG vaccination among other factors. Interferon-Gamma Release Assays (IGRAs) are more accurate for screening but are expensive and usually not available in primary care centers. In our sample, PPD tests were done at arrival in primary attention centers, whereas IGRAs were done in the first visit to the hospital in selected cases (32 patients, 16%) to confirm diagnosis. Despite the advantage of not requiring a second visit for interpretation, some patients were lost to follow-up before the IGRA results were available. 

Our results also suggest that parasitosis are frequent in this population, in line with previous studies [10,11,14,16]. Cost-effectiveness of the screening of parasitic infections is also controversial. Although the potential for overdiagnosis when using serology limits their usefulness according to some authors, others rely on the benefits of early treatment to prevent long-term sequelae [14]. From our point of view, in the unique population of UM, overtreatment may be less harmful than an underdiagnosis, considering the long-term sequelae and scarce access to health care systems. 

The optimal management of this population in terms of immunizations is also unclear. Vaccination cards are unavailable, and no information can be obtained from families in the specific case of UM. Serologies may help for adjusting the schedule, but the cost-effectiveness of this approach is questionable, and measurement of vaccine response is usually not available for many immunizations in routine clinical practices. To ensure protection is crucial, from both an individual but also from a public health perspective, as these children and adolescents may spend many months in institutions, often overcrowded, until their legal status is resolved. Protection against meningococcal disease, hepatitis, varicella, measles, rubella and mumps should be acquired as soon as possible. Our results demonstrate that vaccination coverage is good for most vaccine-preventable diseases, except for measles and hepatitis B. Performing serologies may help to adjust schedules, avoiding unnecessary shots and prioritizing the administration of meningococcal vaccines and a booster dose of HVB and measles. If administering a booster is enough or a full immunization is required should be addressed in further studies. 

If improving diagnostic performance is important, ensuring treatment adherence and linkage to care in this population is an urgent need. In a highly transmissible disease such as TB, a lost-to-follow-up rate above 40% is unacceptable. The arrival of new, ultra-short regimens for TB treatment is encouraging, but ending global TB epidemics will require a specific effort in adolescents, with the design of targeted interventions and the development of multidiscipline units, including cultural and probably also peer support and all the innovative technologies available. Facilitating same day treatment initiation, a strategy that has demonstrated success in HIV treatment, should also be considered. In this study, 21% of patients had not even started treatment. Combined pills are important to increase adherence among adolescents, but are usually more expensive and supply shortages are not uncommon. The fact that all minors included in this study were, at least initially, in the institutional protection system is especially worrisome. A less hostile environment, specific training to overcome cultural barriers, together with translators and cultural mediators, need to be part of the attending teams, to ensure that all barriers are addressed. Although the high rate of patients diagnosed with parasitosis that never started treatment may be related to the complexity of acquiring certain antiparasitic drugs not available in Spain (accessible only via the pharmacy at the Ministry of Health), this is not the case for tuberculostatic drugs.

Our study has several limitations. The characteristics of our unit and references for tropical pediatric diseases limit our ability to generalize results, as many patients are referred because of symptoms and/or abnormal findings during health examinations at primary care or asylum seekers’ centers, including a positive PPD. This fact explains the higher prevalence of LTBI found compared to other studies [1,2,3], and may have influenced the prevalence of certain parasitosis, as some patients were referred because of unexplained eosinophilia. The retrospective design impairs us from tailoring the study protocol, which did not contemplate the screening of other infections such us sexually transmitted diseases. With limited education and skills, unaccompanied children are extremely vulnerable socioeconomically, frequently exposing themselves to sexual and reproductive health risks [29]. However, the evidence is surprisingly scarce, and there is controversy regarding the need for STD screening in this population. Further studies will be needed to answer these questions.

## 5. Conclusions

The vulnerable population of migrating unaccompanied minors is also facing the effects of the COVID-19 pandemic on health care access. The pandemic has highlighted the importance of adequate access to health in this population. Optimizing the management and improving linkage to care is an urgent need to diagnose and treat infectious diseases such as tuberculosis and parasitosis. Addressing the socio-economic conditions that lead to migration should be enforced to prevent the thriving of child migration. 

## Figures and Tables

**Figure 1 tropicalmed-08-00028-f001:**
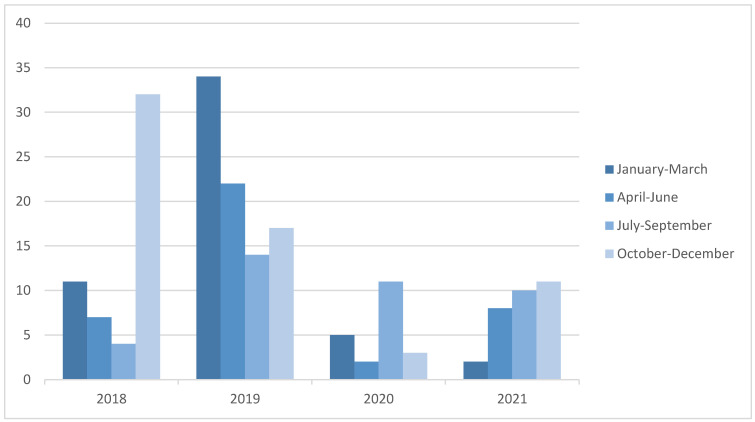
Number of unaccompanied minors screened by year and trimester.

**Table 1 tropicalmed-08-00028-t001:** Global descriptive, pre-pandemic years (2018–2019) and pandemic years (2020–2021).

	Global (N = 192)	Pre-Pandemic 2018–2019 (N = 140)	Pandemic 2020–2021 (N = 52)	*p*
** Demographic **				
**Male sex**	186 (96.9%)	134 (95.7%)	52 (100%)	
**Age** (years) *	16.8 (0.8)	16.75 (0.8)	17.02 (0.9)	
**Weight** (kg) *	64.8 (9.5)	64.6 (9.8)	65.2 (8.9)	
**BMI** (kg/m2) *	21.52 (2.9)	21.6 (3)	21.3 (2.5)	
<18	4 (2.1%)	2 (1.4%)	2 (3.8%)	
25–29	18 (9.4%)	13 (9.2%)	5 (9.6 %)	
>29	2 (1%)	2 (1.4%)	0	
**Reason**(**s**) **for referral**				
Positive PPD	143 (74.5%)	100 (71.4%)	43 (82.7%)	
Eosinophilia	40 (20.8%)	32 (22.7%)	8 (15.4%)	
Imported pathology screening	25 (13%)	21 (15%)	4 (7.7%)	
**Origin region**				
North Africa	113 (58.9%)	88 (62.4%)	25 (48.1%)	
Morocco	109 (56.8%)	86 (61.4%)	23 (44.2%)	0.024
Algeria	4 (2.1%)	2 (1.4%)	2 (3.8%)	
Other regions of Africa	78 (40.6%)	51 (36%)	27 (51.9%)	
Asia (Bangladeshi)	1 (0.5%)	1 (0.7%)	0	
**Spanish language** (acceptable or fluid)	13 (6.7%)	8 (5.7%)	5 (9.6%)	
** Symptoms **	73 (38%)	44 (31.4%)	29 (55.7%)	
Fever	2 (1%)	0	2 (3.8%)	
Asthenia	7 (3.6%)	5 (3.6%)	2 (3.8%)	
Cough	18 (9.4%)	10 (7.1%)	8 (15.4%)	
Diarrhoea	4 (2.1%)	0	4 (7.7%)	0.005
Pruritus	10 (5.2%)	7 (5%)	3 (5.8%)	
Skin lesions	18 (9.4%)	14 (10%)	4 (7.7%)	
Chest pain	3 (1.6%)	2 (1.4%)	1 (1.9%)	
Abdominal pain	11 (5.7%)	6 (4.3%)	5 (9.6%)	
** Diagnosis **				
**Tuberculosis Screening**				
Positive PPD test	143 (74.5%)	100 (71.4%)	43 (82.7%)	
Positive IGRA test	21 (65.6%)	11 (61.1%)	10 (71.4%)	
LTBI diagnosis	140 (72.9%)	97 (69.3%)	43 (82.7%)	
Active TB disease diagnosis	3 (1.57%)	3 (2.1%)	0	
**Viral hepatitis**				
HBeAg+ chronic HBV infection	16 (8.3%)	11 (7.9%)	5 (9.6%)	
HBsAg-phase HBV infection	23 (12%)	22 (15.7%)	1 (1.9%)	
HCV	1 (0.5%)	1(0.7%)	0	
**HIV**	0	0	0	

**Parasites**				
Scabies	5 (2.6%)	3 (2.1%)	2 (3.9%)	
*Giardia lamblia*	9/72 (12.5%)	5 (3.6%)	4 (7.7%)	
*Entamoeba* (*coli* or *dispar*)	5/72 (6.9%)	4 (2.8%)	1 (1.9%)	
*Schistosoma*	34/76 (44.7%)	22 (15.7%)	12 (23.1%)	
*Strongyloides*	7/82 (8.5%)	4 (2.8%)	3 (5.8%)	
*Toxocara*	5/78 (6.4%)	3 (2.1%)	2 (3.8%)	
*B. hominis*	45/72 (62.5%)	32 (22.9%)	13 (25%)	
*D. fragilis*	16/72 (22.5%)	11 (7.9%)	5 (9.6%)	
*B. hominis + D. fragilis*	13/72 (18%)	8 (5.7%)	5 (9.6%)	
*Endolimax nana*	6/72 (9.2%)	3 (2.1%)	3 (5.8%)	
*Taenia* spp.	1/72 (0.5%)	1 (0.7%)	0	
*Ascaris*	4 (2.1%)	3 (2.1%)	1 (1.9%)	
** Immunizations **				
Measles	108/158 (68.4%)	85/114 (74.6%)	23/44 (52.3%)	
Rubella	139/150 (92.7%)	102/108 (94.4%)	37/42 (88%)	
Parotiditis	132/142 (93%)	96/101 (95%)	36/41 (87.8%)	
Varicella	140/163 (85.9%)	105/119 (88.2%)	35/44 (79.5%)	
HAV	128/157 (81.5%)	93/113 (82.3%)	35/44 (79.5%)	
HBV	59/154 (38.3%)	47/108 (43.5%)	19/46 (41.3%)	

* Mean (SD).

## Data Availability

Not applicable.

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
