# Peer review of "Impact of Coronavirus Pandemic on Tuberculosis and Other Imported Diseases Screening among Migrant Minors in Spain"

_tropicalmed, 2022, doi:10.3390/tropicalmed8010028_

Round 1

Reviewer 1 Report

Title:
Title is too long. General recommendations are, ten to twelve appropriately chosen words.

Introduction:
Acknowledge the research work done by others on the topic.

Objectives:
Clearly describe General and Specific objectives of the study.

Methods:

·        This section needs improvement.

·        Include the list of imported diseases, for which the study subjects were screened, and what tests were done.

·        Purified protein derivative PPD and Interferon Gamma Assay (IGRA) were used for screening of Tuberculosis, describe procedures and how tests were conducted.

·        Three patients were diagnosed with active tuberculosis. Please describe how active TB was diagnosed

Results:
The results section should simply state findings without interpretation (149 - blood tests were performed in all patients is a part of methods)

Discussion:
Discuss the important findings of the study and compare your findings with similar studies.

Author Response

Dear reviewer, Thank you very much for your comments. We have incorporated the recommendations mentioned in the manuscript.

Reviewer 2 Report

1. The related and updated manuscript should be added in introduction part.

2. Author should be added the CVOID-19 situation of your study size in introduction part.

3. Author should be coded the reference of diagnosis procedure.

4. The discussion part must be revised by the update and related manuscripts.

Author Response

(The authors gave the same response as above.)

Round 2

Reviewer 2 Report

Accept for publiction